# GABA quantification in human anterior cingulate cortex

**Jan Weis** [1]\*, **Jonas Persson** [2], **Andreas Frick** [3], **Fredrik Åhs** [4], **Maarten Versluis** [5], **Daniel Alamidi** [6]

**1** Department of Medical Physics, Uppsala University Hospital, Uppsala, Sweden, **2** Department of Neuroscience, Uppsala University, Uppsala, Sweden, **3** Department of Neuroscience, The Beijer Laboratory, Uppsala University, Uppsala, Sweden, **4** Department of Psychology and Social Work, Mid Sweden University, Östersund, Sweden, **5** Philips Healthcare, Best, Amsterdam, The Netherlands, **6** Philips, Stockholm, Sweden

\* jan.weis@radiol.uu.se

**Data Availability Statement:** All relevant data are within the manuscript and its Supporting Information files.

**Funding:** J.P. and A. F. were supported by the Swedish Research Council, the Swedish Brain

## Abstract

γ-Aminobutyric acid (GABA) is a primary inhibitory neurotransmitter in the human brain. It has been shown that altered GABA concentration plays an important role in a variety of psychiatric and neurological disorders. The main purpose of this study was to propose a combination of PRESS and MEGA-PRESS acquisitions for absolute GABA quantification and to compare GABA estimations obtained using total choline (tCho), total creatine (tCr), and total N-acetyl aspartate (tNAA) as the internal concentration references with water referenced quantification. The second aim was to demonstrate the fitting approach of MEGA-PRESS spectra with QuasarX algorithm using a basis set of GABA, glutamate, glutamine, and NAA in vitro spectra. Thirteen volunteers were scanned with the MEGA-PRESS sequence at 3T. Interleaved water referencing was used for quantification, $B_0$ drift correction and to update the carrier frequency of RF pulses in real time. Reference metabolite concentrations were acquired using a PRESS sequence with short TE (30 ms) and long TR (5000 ms). Absolute concentration were corrected for cerebrospinal fluid, gray and white matter water fractions and relaxation effects. Water referenced GABA estimations were significantly higher compared to the values obtained by metabolite references. We conclude that QuasarX algorithm together with the basis set of in vitro spectra improves reliability of GABA+ fitting. The proposed GABA quantification method with PRESS and MEGA-PRESS acquisitions enables the utilization of tCho, tCr, and tNAA as internal concentration references. The use of different concentration references have a good potential to improve the reliability of GABA estimation.

## Introduction

Two important neurotransmitters in the mammalian brain are glutamate (Glu) and γ-aminobutyric acid (GABA). It should be noted, that both these compounds are not only neurotransmitters. They have other metabolic functions as well. In fact, only a small portion of the Glu

Foundation, and the Swedish Foundation for Humanities and Social Sciences (Riksbankens jubileumsfond). A.F. was supported by The Kjell and Märta Beijer Foundation and J.P. was supported by a postdoctoral grant from the Swedish Brain Foundation. The funders provided support in the form of salaries for authors [J.P., A. F.], but did not have any additional role in the study design, data collection and analysis, decision to publish, or preparation of the manuscript. The specific roles of these authors are articulated in the 'author contributions' section.

**Competing interests:** Our co-authors D.A. and M. V. declare no conflict of interests in respect of their commercial affiliation in the Philips Healthcare. This does not alter our adherence to PLOS ONE policies on sharing data and materials.

and GABA are neurotransmitters. Glu is the major excitatory transmitter in the brain [1]. Glu serves as a metabolic precursor for GABA which is the primary inhibitory neurotransmitter. GABA plays a crucial role in shaping and regulating neuronal activity [2]. Changes in GABA concentrations have been associated with a variety of neuropsychiatric disorders, such as depression, anxiety, epilepsy, schizophrenia, ADHD, chronic pain, etc. [3,4].

Glu can be quantified with good accuracy by short echo time (TE < 40 ms) magnetic resonance spectroscopy using 3, 4 and 7T scanners [5–7]. However, the quantification of GABA is challenging because GABA spectral lines centered at 1.89, 2.28 and 3.01 ppm are overlapped with the strong signals of total creatine (tCr) (creatine and phosphocreatine), Glu, glutamine (Gln), total N-acetylaspartate (tNAA) (NAA and N-acetylaspartylglutamate (NAAG)), macromolecules (MM) and others. The most widely used approach for GABA detection at 3T is a J-difference Mescher-Garwood (MEGA) spectral editing technique incorporated within a point resolved spectroscopy (PRESS) sequence [8]. MEGA-PRESS exploits the scalar (J) coupling between GABA C4 protons ($^4CH_2$) at 3.01 ppm and C3 protons ($^3CH_2$) at 1.89 ppm. J-difference editing involves the acquisition of two spectra measured with TE 68 ms. The first (ON) spectrum is acquired by applying a pair of frequency-selective GABA-editing RF pulses (center frequency 1.89 ppm). The second (OFF) spectrum is measured with editing frequency-selective pulses at center frequency 7.46 ppm, which are not expected to have an impact on the spectrum. It should be noted that MM resonances at ~1.7 ppm are also inverted by editing frequency-selective RF pulses and coupled to MM protons at 3 ppm. These co-edited MM signals overlap with GABA C4 resonances. The resulting GABA peak is therefore referred to as GABA+ to point out the summation of GABA with MM signals. Recent studies showed that approximately 50% of GABA+ intensity originates from MM [3].

Cerebral GABA content is most often expressed as a spectral intensity ratio of GABA+/tCr. A disadvantage of such an approach is the fact that it is difficult to determine whether the alteration is caused by the numerator, denominator, or both. A suitable example is the tCr concentration that is subject to change in patients with schizophrenia, Alzheimer's and Parkinsson diseases [9,10]. This problem can be minimized by the evaluation of GABA+ spectral intensity ratio to the intensity of other metabolites or to the intensity of the water. The alternative to the unitless spectral intensity ratio is absolute quantification. The most common method for absolute GABA quantification utilizes tissue water as an internal concentration reference [11–13]. However, the quantification is not straightforward because the brain water originates from three tissue compartments: cerebrospinal fluid (CSF), gray (GM) and white matter (WM). Each compartment has different MR-visible water fraction and is weighted by different $T_1$ and $T_2$ relaxation times.

Recently, GABA estimation using tCr as an internal concentration reference was suggested as an alternative to water reference [14,15]. This approach benefits from the fact, that partial volume and relaxation corrections are unnecessary because metabolites originate only from GM and WM compartments and the relaxation times of metabolites are approximately equal in both compartments. Grewal et al [15] assumed tCr to be 7.1 mM. This concentration was estimated for WM using water referenced spectroscopic imaging (PRESS, TR/TE 1500/135 ms) [16]. Similarly, Bhattacharyya et al [14] applied the value 9.22 mM measured by single-voxel PRESS technique (TR/TE 2700/68 ms). A short TR and long TE caused these approaches to be sensitive to the inaccuracies of tCr and water relaxation times in WM, GM, and CSF.

The goal of this study was threefold: 1) to demonstrate a new fitting approach of MEGA-PRESS spectra using QuasarX algorithm as implemented in jMRUI 6.0 software package [17]. A basis set of GABA, Glu, Gln, and NAA in vitro spectra were measured for this purpose; 2) to quantify GABA using water as the internal concentration reference; and 3) to quantify GABA using total choline (tCho: free choline, phosphocholine, and glycerophosphocholine), tCr,

tNAA as the internal concentration references. Contrary to previous studies [14,15], the reliability of reference tCho, tCr, and tNAA concentrations was improved by using a PRESS sequence with short TE (30 ms), long TR (5000 ms) together with partial volume and relaxation corrections for WM, GM and CSF content in each voxel. The overarching aim is to contribute to the improvement of the GABA quantification methodology.

## Material and methods

### Study population

In total, thirteen volunteers (6 females and 7 males) were recruited. The volunteers underwent PRESS and MEGA-PRESS measurements. Mean age of the participants was 37±10 years (range: 24–61). All volunteers were healthy without any history of psychiatric or neurological disorders. Ethical approvals were obtained from local Institutional Review Boards and written informed consent was obtained from each participant.

### Phantoms

Four phantoms were produced according to the guidelines for the LCModel's model spectra [18]. The phantoms contained aqueous solutions of GABA (200 mM), Glu (100 mM), Gln (100 mM), and NAA (50 mM). Each phantom contained a single metabolite. Aqueous solutions were prepared using a phosphate buffer consisting of 72 mM $K_2HPO_4$, 28 mM $KH_2PO_4$, 1 g/L $NaN_3$, and 1 mM sodium trimethylsilyl propanesulfonate (DSS). Solution's pH was adjusted to 7.2. The chemicals were purchased from Sigma-Aldrich AB (Stockholm, Sweden).

### MRI and MRS acquisition protocols

All experiments were performed on 3T scanner (Achieva dStream, Philips Healthcare, Best, The Netherlands). The data were acquired with a 32 channel receiver head coil. Whole brain 3D $T_1$-weighted turbo FFE images (TR/TE 8/3.8 ms, isotropic resolution 1x1x1 $mm^3$) were acquired to guide the positioning of the voxel in the anterior cingulate cortex (ACC) (Fig 1). The GABA spectra were acquired with a MEGA-PRESS sequence using the following parameters: voxel size 4x4x2 $cm^3$, TR/TE 2000/68 ms, 320 alternating ON-OFF spectra, 14 ms GABA-editing RF pulses at 1.9 (ON) and 7.5 (OFF) ppm, spectral bandwidth 2000 Hz, 1024 time domain data points, and 40 blocks. Each block started with the acquisition of one unsuppressed reference water spectral

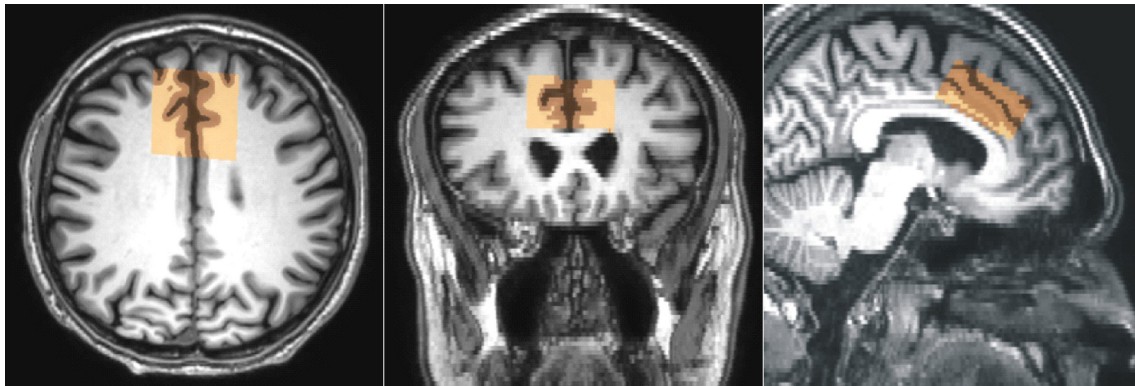

**Fig 1. Spectroscopy voxel position.** Representative voxel (4x4x2 $cm^3$) placement in anterior cingulate cortex and the results of partial volume segmentation of CSF, GM and WM (brown). Transversal images (reconstructed pixel size 0.47x0.47 $mm^2$, slice thickness 2 mm) were used for segmentation and for reconstruction of coronal and sagittal slices.

line followed by four pairs of water suppressed ON-OFF spectra acquired with 4-step phase-cycling. The unsuppressed water signal was used as the internal concentration reference, for eddy current corrections, for $B_0$ drift correction and for updating the carrier frequency of RF pulses in real time. Reference metabolite concentrations were measured using the standard PRESS sequence (TR/TE 5000/30 ms, spectral bandwidth 2000 Hz, 1024 data points, 32 averages, 16 phase cycle steps) with the same voxel size and position. Two dummy excitations were followed by 16 non-water-suppressed and 32 water-suppressed scans. MEGA-PRESS and PRESS sequence performed water suppression by two selective RF pulses and spoiler gradients. Suppression was accomplished by adjusting the flip angle of the second RF pulse such that the longitudinal magnetization of the water signal was minimal at the time of the first excitation RF pulse.

Reference frequency of the PRESS pulses was centered at tCr/GABA (~3 ppm) in water-suppressed scans. Frequency offset was centered at water in scans without water suppression, ie, there was no chemical shift displacement between the water and GABA PRESS-boxes. Chemical shift displacement between GABA and tNAA (2 ppm) PRESS boxes was approximately 2.2, 3.7, and 1.9 mm in left-right (90˚ RF pulse, slice thickness 40 mm), ~feet-head (180˚ RF pulse, slice thickness 40 mm), and ~anterior-posterior (180˚ RF pulse, slice thickness 20 mm) directions, respectively. Chemical shift displacement (absolute value) between GABA and tCho (~3.2 ppm) PRESS boxes was lower by factor 5 compare to displacement between tCr/GABA and tNAA PRESS boxes. It should be noted that 90˚ RF-pulse with broader bandwidth (BW) produces lower chemical shift displacement than 180˚ RF-pulses with narrow BW.

MEGA-PRESS spectra of GABA, Glu, Gln, and NAA aqueous solutions were measured a few hours after preparation. The voxel size was 3x3x3 cm$^3$ and temperature was kept at 22˚C (room temperature) during the acquisition. All other MEGA-PRESS parameters were identical to the in vivo experiments.

## Post processing and quantification

Reconstructed brain images (matrix 512x512, pixel size 0.47 mm, slice thickness 2 mm) were used for GM, WM, and CSF segmentation (Fig 1). Segmentation was performed by using the automated segmentation tool (FAST) [19]. A binary mask of the water PRESS box was created using the SVMask tool (Philips Healthcare, Michael Schär).

MEGA-PRESS spectra were processed with jMRUI 6.0 software package [17]. Each spectrum was zero filled to 8192 points and the residual water was removed by Hankel-Lanczos Singular Value Decomposition (HLSVD) filter. No apodization of the FIDs was applied in this study. All in vivo OFF spectra were averaged and tCho, tCr, and tNAA singlets were fitted using AMARES algorithm (Fig 2). The zero-order phase correction was estimated by AMARES. The in vivo difference (ON-OFF) spectra were fitted by the QuasarX algorithm (QUEST with new constrains and shape peak selection). This time-domain algorithm uses prior knowledge obtained either theoretically by quantum mechanics, or by measuring in vitro aqueous metabolite solutions (Fig 3). Non-linear least-squares algorithm fits a weighted combination of metabolite FIDs to the considered in vivo FID. Our basis set of FID signals was obtained from MEGA-PRESS spectra of GABA (GABA+), Glu, Gln, NAA and NAAG aqueous solutions (Fig 3). NAAG spectrum was approximated by shifted NAA spectrum, with the main peak shifted to 2.045 ppm from 2.01 ppm [18]. GABA+ spectrum was made by modifying GABA spectrum. Contribution of MM signals to GABA was empirically simulated by adding the Lorentzian line (linewidth 5 Hz) to the central peak of pseudo-triplet at 3 ppm. The amplitude was adjusted to be about ~10% higher compare to outer two peaks (Fig 3). Gaussian line shapes were used to fit GABA+ spectral lines. The in vivo difference spectra were first averaged and then fitted by the QuasarX algorithm (Fig 4). The zero-order phase correction was estimated by QuasarX.

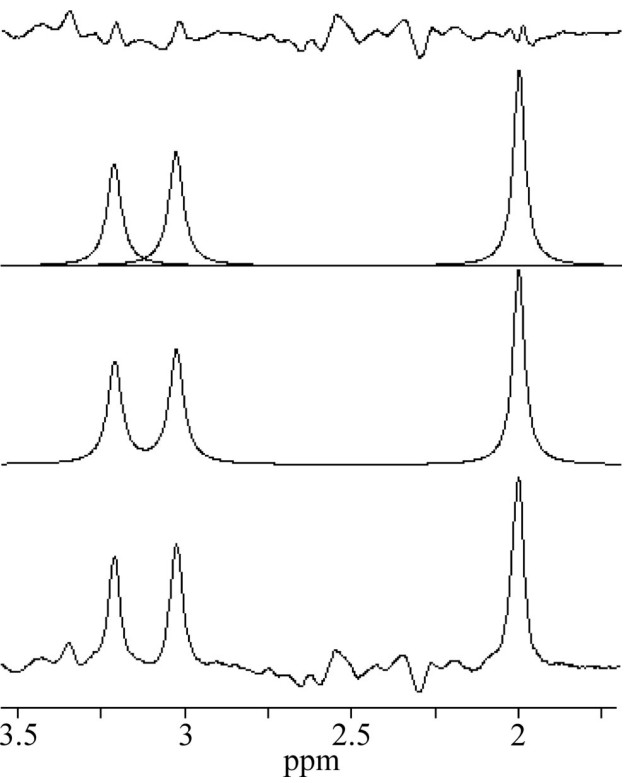

**Fig 2. An example of averaged in vivo OFF spectrum.** AMARES fits of the tCho, tCr, and tNAA singlets, and residue.

The unsuppressed water signal was fitted by Hankel-Lanczos Squares Singular Value Decomposition (HLSSVD) algorithm. The AMARES and QuasarX algorithms provide the Cramér-Rao lower bound (CRLB) standard deviation (CRSD). The fitting error was computed as the percentage ratio of CRSD to the FID's amplitude. Water scaled GABA concentration in relation to wet weight tissue (mol/kg) was computed according to the equation:

$$C_{GABA} = \frac{I_{GABA}}{I_{H2O}} \times \frac{2}{N_{GABA}} \times \frac{1}{R_{GABA}} \times W_{conc} \times \frac{MM_{cor}}{eff_{GABA}} \tag{1}$$

where $I_{GABA}$ is the GABA+ spectral intensity at ~3 ppm, $I_{H2O}$ is intensity of reference water line, $N_{GABA} = 2$ is the number of protons contributing to $I_{GABA}$ resonance, $R_{GABA}$ is the GABA attenuation factor, $MM_{cor} = 0.5$ is a macromolecule correction factor [11,20,21], and $eff_{GABA} = 0.5$ is the editing efficiency [22]. $W_{conc}$ is the reference water concentration corrected for partial volume and relaxation effects [16,23,24]:

$$W_{conc} = \frac{W_{H2O}(f_{GM}R_{H2O-GM} + f_{WM}R_{H2O-WM} + f_{CSF}R_{H2O-CSF})}{(1 - f_{CSF})} \tag{2}$$

and

$$f_x = \frac{c_x \vartheta_x}{0.82\vartheta_{GM} + 0.7\vartheta_{WM} + 0.99\vartheta_{CSF}} \tag{3}$$

where $W_{H2O}$ is the molal concentration of pure water (55.51 mol/kg), $f_x$ is the mole fraction of water in the voxel's GM, WM and CSF, $\vartheta_x$ is the GM, WM and CSF volume fractions and $c_x$ is the relative density of MR visible water in GM (0.82) WM (0.7), and CSF (0.99) [25,26].

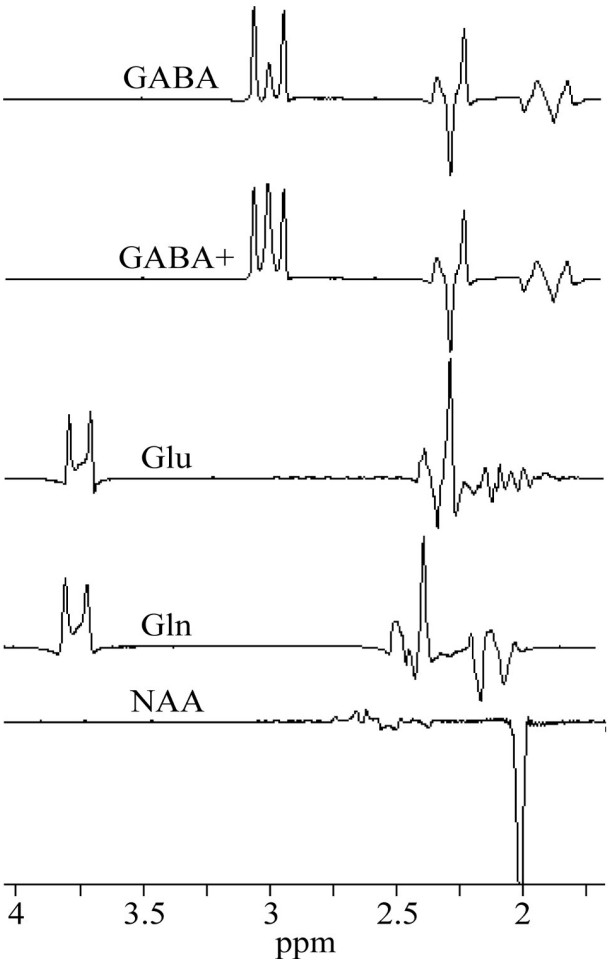

**Fig 3. Basis set of in vitro spectra.** MEGA-PRESS spectra of GABA, Gln, Glu, and NAA aqueous solutions, and simulated GABA+.

$R_{H2O-GM}$, $R_{H2O-WM}$, and $R_{H2O-CSF}$ are PRESS relaxation attenuation factors R = exp(-TE/$T_2$)x [1-exp(-TR/$T_1$)] of water in GM, WM, and CSF, respectively. The following relaxation times were used for corrections: GABA ($T_1$ 1310 ms, $T_2$ 88 ms) [27,28], water in GM ($T_1$ 1820 ms, $T_2$ 99 ms), WM ($T_1$ 1084 ms, $T_2$ 69 ms), and CSF ($T_1$ 4163 ms, $T_2$ 503 ms) [29–31].

The GABA concentration was also assessed using tCho, tCr, and tNAA as the internal concentration references. The reference metabolite concentrations $C_{MET}$, Glu and other metabolites were measured by PRESS sequence with long TR (5000 ms) and short TE (30 ms) to minimize the influence of the water and metabolites relaxation effects. Concentrations were estimated by LCModel [18]. Partial volume and relaxation corrections were performed by adjusting LCModel control parameter WCONC according to the Eq 2, i.e. WCONC = $W_{conc}$ [24]. It should be noted that the default LCModel control parameter ATTH2O for water attenuation correction was switched off (ATTH2O = 1) because water relaxation corrections were already performed in Eq 2. The absolute GABA concentration (mol/kg) was estimated according to the formula:

$$C_{GABA} = \frac{I_{GABA}}{I_{MET}} \times \frac{N_{MET}}{N_{GABA}} \times \frac{R_{MET}}{R_{GABA}} \times C_{MET} \times \frac{MM_{cor}}{eff_{GABA}} \qquad (4)$$

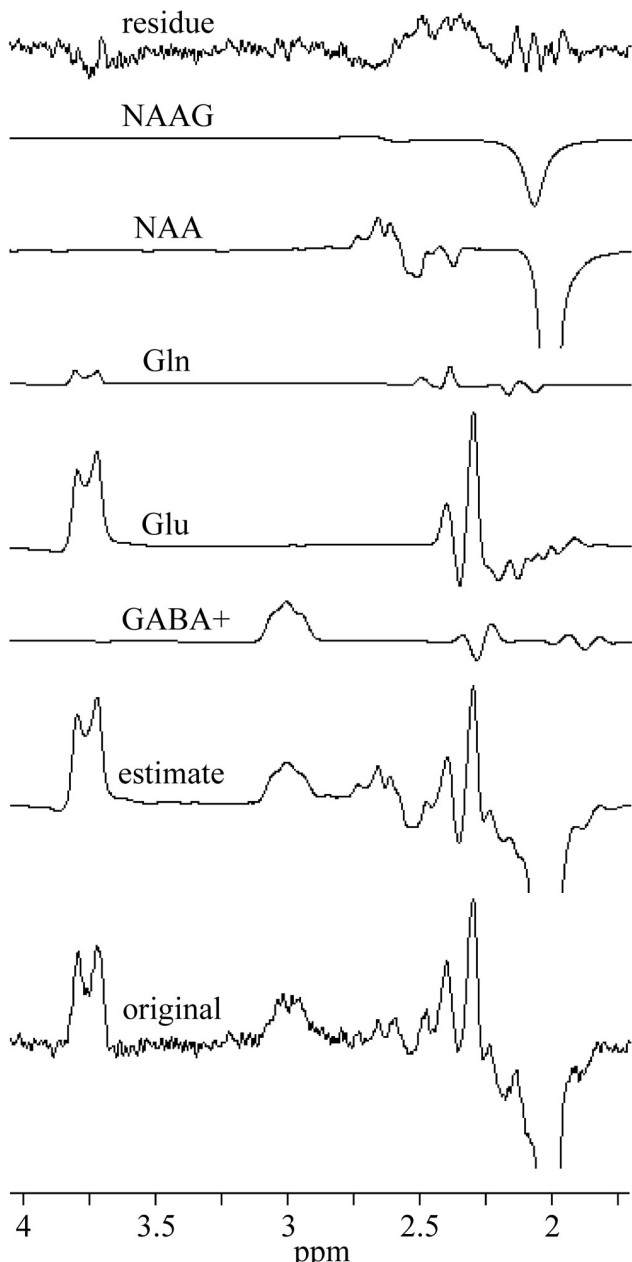

**Fig 4. Representative in vivo MEGA-PRESS spectrum and fits.** The difference spectra were averaged and fitted by the QuasarX algorithm using the basis set spectra shown in Fig 2.

where $I_{MET}$ is the spectral intensity of reference metabolite in OFF spectrum (Fig 3), $N_{MET}$ is the number of protons contributing to $I_{MET}$ resonance (9 for tCho, 3 for tCr and tNAA), $R_{MET}$ is the metabolite attenuation factor. $C_{MET}$ is the reference metabolite concentration (mol/kg) of considered volunteer. Mean relaxation times of tCho ($T_1$ 1140 ms, $T_2$ 230 ms), tCr ($T_1$ 1110 ms, $T_2$ 163 ms), and tNAA ($T_1$ 1340 ms, $T_2$ 260 ms) were used in relaxation corrections [32]. It should be noted that only small differences in metabolite relaxation times were found between GM and WM [32,33].

## Statistics

The reported values are given as the mean ± one SD. $P < 0.05$ of a two-tailed Student's t-test was considered statistically significant. The relative variances (variance-to-mean ratio) were expressed in %. The two-tailed F-test was performed to compare variances of mean GABA concentrations obtained by different quantification methods.

## Results

Thirteen volunteers underwent combined PRESS and MEGA-PRESS examinations. All experiments were successful, no spectra had to be discarded. The OFF and difference (ON-OFF) spectra of all volunteers are shown in S1 Fig (Supporting information). Table 1 summarizes the water-scaled metabolite concentrations and CRLBs acquired by the PRESS (TR/TE 5000/30 ms) sequence. The spectra were processed by LCModel. The mean WM, GM, and CSF volume fractions were 52.0 ± 3.5%, 33.2 ± 2.5%, and 14.8 ± 3.5%, respectively. Spectra of GABA, Glu, Gln, and NAA aqueous solutions and simulated GABA+ spectrum are shown in Fig 3. These spectra were used as prior knowledge for fitting the volunteer's MEGA-PRESS spectra using the QuasarX algorithm. Figs 2 and 4 show representative in vivo results. The mean QuasarX fitting error of GABA+ intensity was 1.5 ± 0.2% (range: 1.2–1.8%). The mean AMARES fitting errors of metabolites were 1.0 ± 0.1%, 0.8% ± 0.1%, and 1.2 ± 0.2% for tCr, tNAA, and tCho, respectively. The mean spectral intensity ratios GABA+/tCr, GABA+/tNAA, and GABA+/tCho were 0.070 ± 0.01, 0.052 ± 0.007, and 0.088 ± 0.013, respectively. The absolute GABA concentrations (mmol/kg) were computed according to Eqs 1 and 4 using tissue water (2.57 ±0.26 [2.7]), tCho (1.63±0.22 [3.1]), tCr (1.46 ± 0.19 [2.6]), and tNAA (1.61 ± 0.22 [3.1]) as internal concentration references. The square brackets depict relative variances. Concentrations are visualized in Fig 5. Two-tailed F-tests detected no differences in the variances of GABA concentrations obtained by different methods.

## Discussion

To the best of our knowledge, this is the first study whereby a PRESS sequence with short TE and long TR together with a MEGA-PRESS sequence were used to estimate the absolute GABA concentration. Applied PRESS method improved the precision of individual reference metabolite concentrations and enabled utilization of tCho, tNAA and tCr as internal concentration references at the expense of a relatively short prolongation of the net measurement time (4 minutes in our case). Spectrum processing approach with QuasarX algorithm and the use of different concentration references have a good potential to improve the reliability of GABA estimation.

**Table 1. Water-scaled metabolite concentrations (mmol/kg) and CRLBs (%).**

|  | Concentration | CRLB |
|---|---|---|
| GABA | 2.65 ± 0.44 | 18.62 ± 2.96 |
| Glu | 10.84 ± 0.54 | 5.23 ± 0.44 |
| Glx | 13.03 ± 1.03 | 6.69 ± 0.48 |
| tNAA | 12.37 ± 0.68 | 2.23 ± 0.44 |
| tCr | 9.11 ± 0.57 | 2.0 ± 0.0 |
| tCho | 2.42 ± 0.20 | 2.85 ± 0.38 |
| mI | 6.43 ± 0.51 | 3.92 ± 0.28 |

Concentrations were estimated from the PRESS spectra (TR/TE 5000/30 ms).

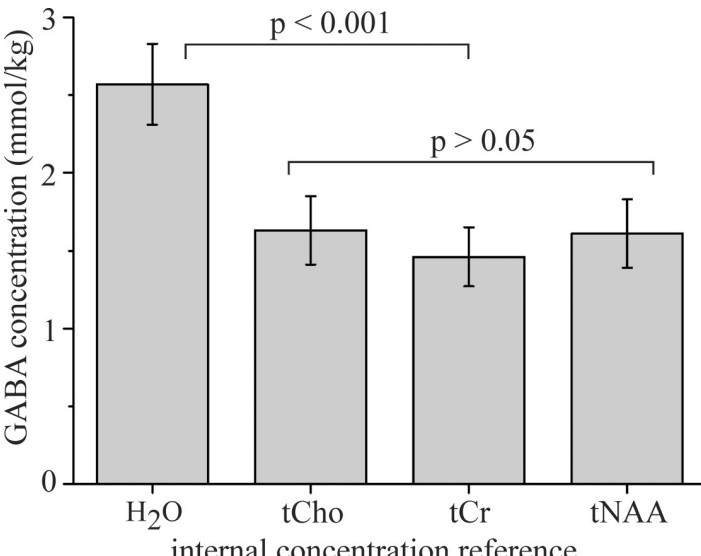

**Fig 5. GABA concentrations.** Concentrations (mmol/kg) were estimated using $H_2O$ (2.57±0.26), tCho (1.63±0.22), tCr (1.46 ± 0.19), and tNAA (1.6±0.22) as internal concentration references.

The anterior cingulate cortex was chosen because this region acts as a central node in the brain and is important for the regulation of advanced brain functions. The water scaled PRESS spectra were used for the individual reference metabolite quantification. The described approach with short TE and long TR together with the partial volume and relaxation corrections is regarded to be the most accurate. This is because errors due to inaccurate relaxation times were minimized. Low CRLBs of Glu, tCho, tCr, and tNAA LCModel fits (Table 1) indicate very good fitting results of the LCModel algorithm. The fitting errors of GABA signals were at the boundary of acceptable reliability (CRLB ~ 20%). However, it should be noted that average over a group of LCModel results can significantly reduce the uncertainty [18]. The concentration estimates of Glu, Glx, tNAA, tCr and tCho are in line with previous studies performed at 3, 4, and 7 Tesla [5,7].

The relative variances of absolute GABA concentrations estimated using $H_2O$, tCho, tCr, and tNAA concentration references show similar dispersion. The comparison of our GABA concentrations with the literature data is not straightforward due to differences in tissue composition and data processing. Cerebral GABA content from ~1 up to 3.7 mM was previously reported [11–15,34,35]. GM/WM ratio is an important issue because GABA content was reported to be from 1.5 to 8.7 times larger in GM relative to WM [14,20,34,36]. Differences in segmentation algorithms, spectrum processing methods, macromolecule correction factor $MM_{cor}$ and accuracy of internal concentration references are also important factors that contribute to the variability of the concentration estimates.

The absolute GABA concentrations estimated using water reference and measured by PRESS and MEGA-PRESS are surprisingly in very good agreement. However, water referenced GABA concentrations were significantly higher than the concentrations estimated with tCho, tCr and tNAA references (Fig 5). The main drawback of water referenced quantification using typical MEGA-PRESS (TR/TE 2000/68 ms) acquisition is the fact that partial volume and relaxation corrections (Eqs 2 and 3) depend on the precision of WM, GM, and CSF segmentation and on the accuracy of nine experimental constants: water fractions and water relaxation times $T_1$, $T_2$ in GM, WM, and CSF. The advantage of GABA quantification using tCr, tCho, and tNAA as the internal concentration references is the fact that partial volume

and relaxation corrections are unnecessary because metabolites originate only from GM and WM compartments and the relaxation times of tCho, tCr, and tNAA are approximately equal in both compartments [32,33]. It should be noted, that the described metabolite reference method is still subject to all sources of error as in water referenced MEGA-PRESS approach because tCho, tCr, and tNAA were quantified from water scaled PRESS spectra (TR/TE 5000/ 30 ms). However, the main difference is in relaxation correction accuracy. Standard water referenced MEGA-PRESS approach with a relatively short TR (2000 ms) and long TE (68 ms) is more susceptible to inaccuracies of relaxation times compare to the proposed metabolite referenced quantification using PRESS with long TR (5000 ms) and short TE (30 ms). It should be noted, that quantification of tCho, tCr, and tNAA can be omitted in comparative studies and the most reliable literature values can be applied instead. Our GABA values can be compared with the concentrations estimated from the water scaled STEAM and SPECIAL spectra measured at 7 T scanners [5,7]. The occipital lobe spectra were measured with long TR and a very short TE (6 ms). High spatial resolution facilitated fitting of the GABA triplet at 2.28 ppm which is uncontaminated by the macromolecules. GABA levels in the range of 1.3–1.6 mmol/ kg were reported. These values were slightly underestimated because partial volume and relaxation corrections were not taken into consideration. Nevertheless, we believe that our metabolite referenced results conform to the most reliable literature values such as the GABA values reported by Mekle et al [5] and Tkac et al [7]. We hypothesize that our water referenced GABA values are overestimated due to inaccuracies in partial volume and relaxation corrections.

## Conclusion

QuasarX algorithm together with the basis set of in vitro spectra improves reliability of GABA + fitting. The proposed GABA quantification method with PRESS and MEGA-PRESS acquisitions enables the utilization of tCho, tCr, and tNAA as internal concentration references. Water referenced GABA estimations were significantly higher compared to the values obtained by metabolite references. The use of different concentration references have a good potential to improve the reliability of GABA estimation.

## Supporting information

**S1 Fig. MEGA-PRESS spectra.** OFF spectra (left) and corresponding difference spectra (right) of all volunteers.
(TIF)

## Author Contributions

**Conceptualization:** Jan Weis, Jonas Persson, Andreas Frick, Fredrik Åhs, Maarten Versluis, Daniel Alamidi.

**Data curation:** Jan Weis, Jonas Persson, Andreas Frick, Daniel Alamidi.

**Formal analysis:** Jan Weis, Jonas Persson, Andreas Frick, Fredrik Åhs, Maarten Versluis, Daniel Alamidi.

**Funding acquisition:** Jonas Persson, Andreas Frick, Fredrik Åhs.

**Methodology:** Jan Weis, Jonas Persson, Andreas Frick, Fredrik Åhs, Maarten Versluis, Daniel Alamidi.

**Project administration:** Andreas Frick, Daniel Alamidi.

**Resources:** Maarten Versluis, Daniel Alamidi.

**Software:** Maarten Versluis, Daniel Alamidi.

**Supervision:** Daniel Alamidi.

**Validation:** Jonas Persson, Fredrik Åhs, Daniel Alamidi.

**Visualization:** Jan Weis.

**Writing – original draft:** Jan Weis.

**Writing – review & editing:** Jan Weis, Jonas Persson, Andreas Frick, Fredrik Åhs, Maarten Versluis, Daniel Alamidi.

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
