## [Decision Letter · Decision Letter 0]

30 Nov 2020

PONE-D-20-30419

GABA quantification in human anterior cingulate cortex

PLOS ONE

Dear Dr. Weis,

Thank you for submitting your manuscript to PLOS ONE. After careful consideration, we feel that it has merit but does not fully meet PLOS ONE’s publication criteria as it currently stands. Therefore, we invite you to submit a revised version of the manuscript that addresses the points raised during the review process.

Please update according to reviewers comments. Especially it would be prudent to update on the biological meaning of 'neurotransmitters', also remove some of the more basic explanations of the editing method, while expanding substantially on the experiment details, as suggested by reviewers.

We look forward to receiving your revised manuscript.

Kind regards,

Peter Lundberg

Academic Editor

PLOS ONE

Journal Requirements:

2. We noted that two co-authors are affiliated with Philips healthcare and that you used materials manufactured by Philips healthcare. Please consult our guidelines on declaring competing interests at https://journals.plos.org/plosone/s/competing-interests and update your competing interests declaration in the submission form as appropriate.

         2. Please also provide an updated Competing Interests Statement declaring this commercial affiliation a      long with any other relevant declarations relating to employment, consultancy, patents, products in development, or marketed products, etc.  

Reviewers' comments:

Reviewer's Responses to Questions

**Comments to the Author**

1. Is the manuscript technically sound, and do the data support the conclusions?

Reviewer #1: Yes

Reviewer #2: Partly

2. Has the statistical analysis been performed appropriately and rigorously? 

Reviewer #1: Yes

Reviewer #2: Yes

3. Have the authors made all data underlying the findings in their manuscript fully available?

Reviewer #1: No

Reviewer #2: Yes

4. Is the manuscript presented in an intelligible fashion and written in standard English?

Reviewer #1: Yes

Reviewer #2: Yes

5. Review Comments to the Author

Reviewer #1: Summary:

In this study, the authors propose a combination of PRESS and MEGA-PRESS acquisitions for absolute GABA quantification. In addition, the authors also demonstrate the QuasarX algorithm implemented in jMRUI used for the purpose of fitting the MEGA-PRESS spectra. This proposed GABA quantification method based on utilization of tCho, tCr, and tNAA as internal concentration references showed good potential for improving the reliability of the GABA estimation. The authors give a thorough background of the field ending in well-defined aims of the study. Furthermore, the authors have been very clear by describing every step carefully in the method section and considering potential explanations for the results in the discussion section. However, my main concern about this manuscript is how the results are presented, thus the figures and tables. If this presentation is improved, in combination with some clarifications to the following points, I think this great work could be published.

Major Comments:

1. Please show all spectra and not just representative spectra. It is easier for the reader to see overall quality of the data acquired in this study by looking at all spectra, than just look at a spectrum chosen by the authors. This point is valid for both Figure 3 and 4, but especially Figure 4.

2. Please further clarify how the zero-order correction was performed. Figure 3 is not very well explained. Did you only use OFF data for the correction? (Line 172)

3. Why are not Table 1 and Table 2 in the same table with all the concentration ratios computed for all reference metabolite (as done with the water reference)? Alternatively, remove Table 2 because it contains 3 numbers that easily could be written in the text.

4. Table 3 is completely redundant because it is containing the same information that is described in Figure 5. Consider removing Table 3 and add these numbers in Figure 5 (maybe over each bar?) or simply just include these in the caption and text.

5. All figures (except Figure 5) are very blurry. If this is a consequence of the submission, disregard this comment, otherwise, increasing the figure quality will substantially improve the overall impression of this work.

Minor Comments:

• Methods: What type of water suppression was used?

• Line 165: Consider rewording sentence “This nonlinear least-squares algorithm fits a time-domain model function, made up from a basis set of in vitro spectra, to in vivo data.”

• Line 172: Something is strange: “The zero-order phase correction of in vivo MEGA-PRESS spectrum was estimated by fitting the tCho, tCr, and tNAA singlets in averaged OFF spectrum using AMARES algorithm (Fig 3).”

• Line 248: what were the p-values?

Reviewer #2: This manuscript describes the acquisition and processing of edited spectra obtained from the anterior cingulate cortices of 13 healthy volunteers. It aims to demonstrate a new approach to fitting edited spectra as well as to quantify GABA using both water and other internal metabolites as concentration references. The overarching aim, although not explicitly stated, seems to be to test whether there is superior reliability of the above approaches compared to the published literature.

The authors have obtained some good quality spectra and have used a method for post-processing which is claimed to improve the precision of measurement.

Introduction:

It is stated that glutamate is involved in EVERY major excitatory brain function (referencing a 2000 review by Meldrum). This is a generalization and is not strictly accurate. There are other excitatory neurotransmitters which may act independently of glutamatergic neurotransmission and which also play important roles (without which the organism concerned will die, which suggests that they are probably playing major roles too). If the authors are referencing a review for the role played by glutamate, a more up to date one (e.g. Zhou and Danbolt 2014 or would be more appropriate as the field has moved since 2000.

Several paragraphs are expended describing the basis of the MEGA-PRESS approach, which is already well documented in the literature and could be removed and replaced with suitable references without losing any understanding.

It is important to note that calling both glutamate and GABA “neurotransmitter” is somewhat misleading. Only a very small fraction of the Glu and GABA in the MR spectrum is actually neurotransmitter while the rest of it is metabolic. Glutamate is the major contributor to transaminase reactions in the brain where it is used to maintain equilibria with other amino acids such as aspartate and alanine as well as to buffer the Krebs cycle. While all the metabolic glutamate can potentially become synaptic glutamate (and potentially thereafter a neurotransmitter) by far the biggest impact on glutamate levels is activity based and changes in the levels can be caused by many different factors. Similarly with GABA, it is now fairly well accepted that the GABA measured by MRS is mostly extrasynaptic. While it can serve as a neurotransmitter the mechanism for this is largely through tonic inhibition rather than synaptic. The point is that both of these compounds as not simply “neurotransmitter” and calling them this is oversimplifying, potentially misleading and something that should not be encouraged in the literature.

While the goals of the study are stated they are to demonstrate that a method works and then to measure two other things. While these could be worthy goals it would be useful to state the purpose of doing them – what is the over-arching thing that the authors are attempting to achieve and what is the context? Are they showing that the method they are demonstrating is better than other approaches? Why are they doing the measurements? It is currently not clear.

Methods:

Please include the acquisition frequency (F1 – e.g. was the spectrum acquired at the GABA resonant frequency?) and whether the location of the voxel shown is at the frequency of the water resonance or F1) if this was different to 4.7 ppm? Was the voxel parcellated at the frequency of the water or the GABA?

What was the water suppression method used? This can have some impact on the integral of the creatine and the NAA due to exchange.

What was the rationale for zero filling the spectra to 8k data points? Does this impact the fitting at all? And were the MEGA-PRESS spectra processed offline? Were the blocks added on the spectrometer or in jMRUI (this may impact signal to noise depending on how it was done).

P12 line 286 – it is stated that the size of the CRLBs reflects the accuracy of the concentration estimates. It should be noted that CRLBs, although often used as a proxy for error, actually only represent a measure of the goodness of fit of the algorithm to the data and as such are not predictors of ground truth. In order to be able to make this statement, the authors would need to know the actual brain GABA concentration.

The water reference vs Cho.Cre reference differences may be an artefact of jMRUI, depending on how the postprocessing of the MEGA-PRESS spectra was done. Subtraction of one spectrum from another in jMRUI can give different noise values and interfere with the scaling. The unsuppressed, reference water spectrum would not be impacted by this. It is not possible to know if this is a problem here as the methods section does not explain how the post processing of the spectra was done.

Minor points:

P12 line 275, Applied PRESS method improved the ACCURACY of .. should be PRECISION instead of accuracy. Please be careful about use of words like “reliability” when you may mean “repeatability”. Suggest to refer to Bartlett & Frost Ultrasound Obstet Gynecol 2008; 31: 466–475 for a useful discussion of these terms and their correct usage.

A paper has recently been published online in Magn Reson Med https://doi.org/10.1002/mrm.28587 which examines the repeatability and reliability of GABA measures in the anterior cingulate cortex

For noting:

The T2s of the relevant macromolecules have recently been published doi.org/10.1002/mrm.28282. and could be included in the calculations. Similarly, a method was shown which enabled one to propagate the estimation errors through the calculation; this led to improvements in precision.

6. PLOS authors have the option to publish the peer review history of their article (what does this mean?). If published, this will include your full peer review and any attached files.

Reviewer #1: No

Reviewer #2: No

---

## [Author Response · Author response to Decision Letter 0]

21 Dec 2020

Response to reviewers

We thank to the academic editor and reviewers for their professional review and constructive criticism.

Academic editor

Please update according to reviewers comments. Especially it would be prudent to update on the biological meaning of 'neurotransmitters', also remove some of the more basic explanations of the editing method, while expanding substantially on the experiment details, as suggested by reviewers.

Response: 

- The first paragraph in “Introduction” (page 3) was rewritten according the hints of the reviewer #2 in point R2.C2.

- Basic explanation of MEGA editing method was removed. Please see point R2.C3.

- The experimental details were expanded following the reviewers comments in points R1.C2, R1.C6, R1.C7, R1.C8, R2.C5, R2.C6, R2.C7 

Reviewer #1

R1.C1

1. Please show all spectra and not just representative spectra. It is easier for the reader to see overall quality of the data acquired in this study by looking at all spectra, than just look at a spectrum chosen by the authors. This point is valid for both Figure 3 and 4, but especially Figure 4.

Response: Done. OFF spectra (Fig. 2) and difference spectra (Fig. 4) of all 13 volunteers are in Supporting information, S1 Fig. This information contains 2-nd sentence in “Results”, page 11. 

R1.C2

2. Please further clarify how the zero-order correction was performed. Figure 3 is not very well explained. Did you only use OFF data for the correction? (Line 172).

Response: We apologize for unclear text. The text in “Post processing and quantification” section (page 8), was improved and figures 2 and 3 have been replaced. Also the text in Fig. 2 and 4 legends was improved. All (160 in our case) OFF spectra were averaged (Fig. 2) as well as all (160) difference (ON-OFF) spectra (Fig. 4). The zero-order phase correction of averaged OFF spectrum was estimated by AMARES algorithm. The same zero-order phase correction algorithm is a part of QuazarX algorithm as well. Zero-order phase correction of difference in vivo MEGA-PRESS spectrum (Fig. 4) was performed by QuazarX. Please see the new text in “Post processing and quantification” section, page 8. 

R1.C3

3. Why are not Table 1 and Table 2 in the same table with all the concentration ratios computed for all reference metabolite (as done with the water reference)? Alternatively, remove Table 2 because it contains 3 numbers that easily could be written in the text.

Response: We agree that Table 2 is redundant. Table 2 was removed and the results were moved to “Results” (page 11) as you suggested. 

R1.C4

4. Table 3 is completely redundant because it is containing the same information that is described in Figure 5. Consider removing Table 3 and add these numbers in Figure 5 (maybe over each bar?) or simply just include these in the caption and text.

Response: Thank you for suggestion. Table 3 was removed. GABA concentrations were added to the end of “Results” section (page 11) as well as to Fig. 5 legend.

R1.C5

5. All figures (except Figure 5) are very blurry. If this is a consequence of the submission, disregard this comment, otherwise, increasing the figure quality will substantially improve the overall impression of this work.

Response: Yes, they are blurry. It is the consequence of magnification and by jpg format used by PlosOne. Submitted original images possess a high quality. 

Minor Comments:

R1.C6

• Methods: What type of water suppression was used?

Response: We used “excitation” water suppression. This kind of water suppression uses two bandwidth selective RF pulses and spoiler gradients. Suppression was accomplished by adjusting the flip angle of the second RF pulse such that the longitudinal magnetization of the water signal was minimal at the time of the first MEGA-PRESS (PRESS) excitation RF pulse. The “MRI and MRS acquisition protocols” section was completed by this information. Please see page 6.

R1.C7

• Line 165: Consider rewording sentence “This nonlinear least-squares algorithm fits a time-domain model function, made up from a basis set of in vitro spectra, to in vivo data.”

Response: This part of manuscript was rewritten. Please see “Post processing and quantification” section, page 8.

R1.C8

• Line 172: Something is strange: “The zero-order phase correction of in vivo MEGA-PRESS spectrum was estimated by fitting the tCho, tCr, and tNAA singlets in averaged OFF spectrum using AMARES algorithm (Fig 3).”

Response: This problem was already solved in our answer R1.C2.

R1.C9

• Line 248: what were the p-values?

Response: We are not sure what you mean because this line contain concentrations and variations. If you mean GABA concentrations then p-values are in Fig. 5. We performed F-test described here:

https://www.statisticshowto.datasciencecentral.com/probability-and-statistics/hypothesis-testing/f-test/

F_critical_table for alpha 0.025:

https://www.statisticshowto.datasciencecentral.com/tables/f-table/

This F-test of variances does not produce the p-values. F-test evaluates two hypothesis: 

Null hypothesis: Ho = the two populations of variances are equal 

Alternative hypothesis: H1 = the two populations of variances are unequal

The input values are variances (SD^2) of two concentrations and degree of freedom of two considered data sets (concentrations). F_critical is then computed and with the help of the table of critical values is null hypothesis either rejected or not.

Reviewer #2

R2.C1: It is stated that glutamate is involved in EVERY major excitatory brain function (referencing a 2000 review by Meldrum). This is a generalization and is not strictly accurate. There are other excitatory neurotransmitters which may act independently of glutamatergic neurotransmission and which also play important roles (without which the organism concerned will die, which suggests that they are probably playing major roles too). If the authors are referencing a review for the role played by glutamate, a more up to date one (e.g. Zhou and Danbolt 2014 or would be more appropriate as the field has moved since 2000.

Response: Thank you for explanation. Reference 1 was replaced by the updated review of Zhou Y, Danbolt NC. J Neural Transm 2014; 121:799-817. The word “every” was deleted and this sentence was rewritten. Please see the first paragraph in “Introduction”, page 3.

R2.C2: It is important to note that calling both glutamate and GABA “neurotransmitter” is somewhat misleading. Only a very small fraction of the Glu and GABA in the MR spectrum is actually neurotransmitter while the rest of it is metabolic. Glutamate is the major contributor to transaminase reactions in the brain where it is used to maintain equilibria with other amino acids such as aspartate and alanine as well as to buffer the Krebs cycle. While all the metabolic glutamate can potentially become synaptic glutamate (and potentially thereafter a neurotransmitter) by far the biggest impact on glutamate levels is activity based and changes in the levels can be caused by many different factors. Similarly with GABA, it is now fairly well accepted that the GABA measured by MRS is mostly extrasynaptic. While it can serve as a neurotransmitter the mechanism for this is largely through tonic inhibition rather than synaptic. The point is that both of these compounds as not simply “neurotransmitter” and calling them this is oversimplifying, potentially misleading and something that should not be encouraged in the literature.

Response: We are grateful for these comments. We did not know it. The first paragraph in “Introduction” (page 3) was rewritten.

R2.C3: Several paragraphs are expended describing the basis of the MEGA-PRESS approach, which is already well documented in the literature and could be removed and replaced with suitable references without losing any understanding.

Response: Six lines in second paragraph of “Introduction” (page 3) were removed. We think that ON, OFF, and GABA+ terms need to be introduced in this paragraph, because they are using in “Material and Methods” sections.

R2.C4: While the goals of the study are stated they are to demonstrate that a method works and then to measure two other things. While these could be worthy goals it would be useful to state the purpose of doing them – what is the over-arching thing that the authors are attempting to achieve and what is the context? Are they showing that the method they are demonstrating is better than other approaches? Why are they doing the measurements? It is currently not clear.

Response: The overarching goal of this study is to contribute to the improvement of the GABA quantification methodology. This sentence was added to the end of the “Introduction”, page 5.

R2.C5: Methods:

Please include the acquisition frequency (F1 – e.g. was the spectrum acquired at the GABA resonant frequency?) and whether the location of the voxel shown is at the frequency of the water resonance or F1) if this was different to 4.7 ppm? Was the voxel parcellated at the frequency of the water or the GABA?

Response: Philips scanners visualize two voxels during voxel position planning. The first voxel is “PlanScan metabolite” and the second is “Shifted metabolite”. User defines which spectral line (or position of line in ppm) represents “PlanScan metabolite” voxel and “Shifted metabolite” voxel. We used 3 ppm position, i.e. Cr/GABA line for PlanScan metabolite. This GABA resonance frequency was use as the reference frequency of the PRESS excitation pulses during acquisition with water suppression. Philips approach is that positions of the water voxel (concentration reference, unsuppressed water line) and “PlanScan metabolite” voxel (tCr/GABA at 3 ppm in our case) are identical. It is achieved by changing the reference frequency of the PRESS excitation pulses during acquisition of unsuppressed water line and during acquisition with water suppression. Voxel shown in Fig. 1 represents position of both the water and “PlanScan metabolite” voxel (GABA/tCr, 3 ppm) voxel. There was no chemical shift displacement between GABA (3 ppm) and water PRESS boxes, very low displacement between tCho and water/GABA PRESS boxes, and somewhat higher (but still acceptable low) displacement between tNAA and water/GABA PRESS boxes. SVMask tool (Philips Healthcare, Michael Schär) mentioned in the 1-st paragraph of “Post processing and quantification” section (page 8) computes binary mask for water = Cr = GABA PRESS boxes (Fig. 1). The chemical shift displacement of the Cho, and NAA voxels has very low impact on GABA quantitation with tNAA, tCho, references because of small differences in displacement between tNAA, tCho, and tCr /GABA PRESS boxes. This item is explained in the penultimate paragraph in “MRI and MRS acquisition protocols” section, page 7.

R2.C6: What was the water suppression method used? This can have some impact on the integral of the creatine and the NAA due to exchange.

Response: This question was already answered in point R1.C6.

R2.C7: What was the rationale for zero filling the spectra to 8k data points? Does this impact the fitting at all? And were the MEGA-PRESS spectra processed offline? Were the blocks added on the spectrometer or in jMRUI (this may impact signal to noise depending on how it was done).

Response: The aim of zero filling was to improve digital resolution of the spectra. It improved visualization of the spectra in very short intervals (1.75 and 2.25 ppm) used in Figs. 2-4. It has negligible effect on the fitting. jMRUI can read spectra in Philips SDAT/SPAR format. All spectra (320 in our case) were exported from the scanner. This spectra were already rearranged by the scanner. The first half of the spectra (160 in our case) are ON spectra and second 160 spectra are OFF spectra multiplied by -1. Difference (ON-OFF) spectra (Fig. 3 and 4) are then computed by summation of all 320 spectra. OFF spectra (Fig. 2) were computed by summation of the second half of the spectra. It should be noted that LCModel is unable to read our SDAT/SPAR data. LCModel assumes the alternating spectra ON, OFF, ON, OFF, etc. To circumvent this, we measured our own basis set (Fig. 3) and processed the spectra using jMRUI. 

R2.C8: P12 line 286 – it is stated that the size of the CRLBs reflects the accuracy of the concentration estimates. It should be noted that CRLBs, although often used as a proxy for error, actually only represent a measure of the goodness of fit of the algorithm to the data and as such are not predictors of ground truth. In order to be able to make this statement, the authors would need to know the actual brain GABA concentration.

Response: Thank you for this alert. This sentence was rewritten. Please see second paragraph of “Discussion”, page 13.

R2.C9: The water reference vs Cho.Cre reference differences may be an artefact of jMRUI, depending on how the postprocessing of the MEGA-PRESS spectra was done.

Response: We are not sure what artifact you have in mind. Our answer to the point R2.C7 contains an additional description of jMRUI processing. Response R2.C5 deals with possible chemical shift displacement errors. We hope that it is clearer now. 

R2.C10: The Subtraction of one spectrum from another in jMRUI can give different noise values and interfere with the scaling. The unsuppressed, reference water spectrum would not be impacted by this. It is not possible to know if this is a problem here as the methods section does not explain how the post processing of the spectra was done.

Response: From S1 Fig (Supporting information) follows that SNR of difference spectra is worse compare to OFF spectrum. It is the consequence of subtraction as you mentioned. However, SNR of difference spectra (Fig. 4, S1 Fig) is still very good taken in to account that no apodization of the FIDs was applied in this study. We think that we used standard description of jMRUI spectrum processing. Our answers to the points R2.C7 and R1.C2 contain an additional description. We hope that it is clearer now. We will be happy to improve spectrum processing section if we know what is unclear. 

R2.C11: P12 line 275, Applied PRESS method improved the ACCURACY of .. should be PRECISION instead of accuracy. 

Response: Corrected. Please see the first paragraph of Discussion, page 12. Thank you. 

Thank you for the recommendation of interesting papers. We will read them.

---

## [Editor Report · Decision Letter 1]

7 Jan 2021

GABA quantification in human anterior cingulate cortex

PONE-D-20-30419R1

Dear Dr. Weis,

We’re pleased to inform you that your manuscript has been judged scientifically suitable for publication and will be formally accepted for publication once it meets all outstanding technical requirements.

Kind regards,

Peter Lundberg

Academic Editor

PLOS ONE
---

## [Editor Report · Acceptance letter]

8 Jan 2021

PONE-D-20-30419R1 

GABA quantification in human anterior cingulate cortex 

Dear Dr. Weis:

I'm pleased to inform you that your manuscript has been deemed suitable for publication in PLOS ONE. Congratulations! Your manuscript is now with our production department. 

Kind regards, 

on behalf of

Professor Peter Lundberg 

Academic Editor

PLOS ONE